# Impact of screening on cervical cancer incidence in England: a time trend analysis

Francesca Pesola, Peter Sasieni

School of Cancer and Pharmaceutical Sciences, Faculty of Life Sciences and Medicine, King's College London, London, UK

**Correspondence to**
Dr Francesca Pesola;
francesca.pesola@kcl.ac.uk

## ABSTRACT

**Objectives** To better model underlying trends in cervical cancer incidence so as to model past trends, to estimate the impact of cervical screening on cervical cancer rates at different ages and to obtain a counterfactual baseline under a no-screening scenario.

**Design** Trend analysis of cancer registry data recorded between 1971 and 2013.

**Setting** England.

**Participants** 132 493 women aged 20–84 with a diagnosis of cervical cancer.

**Outcome measure** Cervical cancer incidence data were modelled using a modified age period cohort model able to capture both increased exposure to human papillomavirus (HPV) as well as changes in the age of exposure to HPV in young cohorts. Observed rates were compared with counterfactual baseline rates under a no-screening scenario to estimate the protective effect of screening.

**Results** Rates of cervical cancer incidence have been decreasing since the introduction of screening but are projected to increase in the future under the current scenario. Between 1988 and 2013, it was estimated that screening had prevented approximately 65 000 cancers. Moreover, in 2013, the age-standardised rate (ASR) estimated under the no-screening scenario (37.9, 95% CI 37.4 to 38.3) was threefold higher among women aged 20–84 than the observed ASR (12.8, 95% CI 12.3 to 13.3). We estimate that the age of first HPV exposure has decreased by about 1 year every decade since the early 1970s (women born in 1955 onwards).

**Conclusions** Our results corroborated the importance of screening in preventing cervical cancer and indicated future rates are dependent on age at HPV exposure. Estimated future rates can be used for healthcare planning while the counterfactual baseline to quantify the impact of HPV vaccination in microsimulations.

## Strengths and limitations of this study

► This is a study on cervical cancer incidence over a 40-year period.
► This is the first study to use a modified age period cohort (APC) model able to capture generational changes in the prevalence of human papillomavirus (HPV) as well as changes in age at first exposure.
► The model allows researchers and policy makers to estimate the effectiveness of screening as well as the effect of HPV vaccination and screening in the future by obtaining a counterfactual baseline.
► Our revised model could be used to study other diseases where age at exposure and intensity of exposure affect future age-specific incidence (eg, lung cancer).
► One limitation, typical of APC-based analyses, is that estimates are dependent on the assumptions of the model.

## INTRODUCTION

It is essential to be able to estimate the future burden of cervical cancer to ensure sufficient resources and services are in place to deal with demands; however, this task is complicated because different risk and preventive factors affect incidence rates of cervical cancer. Specifically, national screening programmes are a main protective factor together with human papillomavirus (HPV) vaccination.[1–3]

Indeed, in England the incidence of cervical cancer has decreased by over a third (from 15.0 to 9.8 per 100 000) since the introduction of the national cervical screening programme in 1988; however, this reduction has slowed in recent years and incidence rates in women under the age of 30 have been increasing.[4] This increase may be driven by greater and earlier exposure to persistent infection of HPV, which has been identified as the main risk factor for cervical cancer.[5–7]

To estimate the future burden of cancer, epidemiologists and medical researchers have implemented age period cohort (APC) models. When exploring trends of cervical cancer, the period effect is a proxy for the effect of screening, while the cohort effect is interpreted as generational changes to HPV exposure, and specifically increased intensity (ie, incidence) in younger cohorts.[8–10] In a recent study, past incidence rates of cervical cancer in the UK were extrapolated to 2035 using an APC model.[11] The overall age-standardised rate (ASR) was predicted to increase by 43% between 2014 and 2035. This increase was observed especially among

women aged 25–49, for whom estimated rates were predicted to be higher by 2035 than they were prior to the introduction of screening. The epidemic predicted in this study is driven by high rates of cervical cancer observed in young women in recent years. Indeed, a standard APC model assumes cohorts with very high rates of cervical cancer at ages 20–29 will continue to have high rates of cancer throughout their lives. We suggest that the timing of the exposure is also important since the risk of cervical cancer was found to be a function of age at first HPV exposure.[12] Hence, the cohort effect needs to capture increased as well as earlier HPV exposure among younger women. Earlier HPV exposure can explain the increased rates of cervical cancer in women in their 20s, but its impact will be minimal by the time these women reach their 40s, 50s and 60s. This is because cervical cancer incidence within a cohort increases sharply between ages 20 and 29 but is relatively flat between ages 40 and 60. The assumption that earlier HPV exposure is cohort-dependent is supported by evidence which shows earlier age at intercourse, a proxy for HPV exposure, in younger cohorts.[13 14]

To explore past trends and estimate future incidence rates of cervical cancer, we developed a modified APC model implicitly able to capture birth cohort changes in HPV exposure both in terms of changes to cumulative exposure and changes to the age at which women are infected. The latter is the innovative feature of our approach. An important output of our model is that it also allows us to estimate rates under a scenario where no screening is available (ie, counterfactual baseline). This counterfactual baseline can be used to quantify the protective effect of screening and the effect of HPV vaccination and screening in future cohorts.[15 16]

## METHODS

### Cases

Data on 132 493 (incidence of) cervical cancers in women aged 20–84 diagnosed between 1971 and 2013 in England were provided by Public Health England. Data were broken down by year of diagnosis and single year of age. The population estimates (women aged 20–84) for 1971–2014 and projections to 2030 were obtained from the Office for National Statistics.[17]

Cancers diagnosed in women aged 24–25 in the most recent years (2010–2013) were excluded from our projections as they tend to be early-stage cancers (Stage IA based on the Fédération Internationale de Gynécologie et d'Obstétrique, FIGO, system) and reflect prevalent disease detected at the first screen.[18 19] Thus, they would require separate modelling. Cancers diagnosed in 2009 for women aged 20–49 were also excluded, as there was a substantial one-off increase in diagnostic screening and symptomatic testing in 2009 as a result of Jade Goody's cancer diagnosis and death.[20 21]

### APC model

APC models, assuming a Poisson distribution of cancer events, are described by the following formula:

$$\lambda(age, period) = g^{-1}[f_A(age) + f_C(cohort) + f_P(period)] \quad (1)$$

where $\lambda$ is the incidence rate as a function of age and calendar year, g is the link function, and $f_A$, $f_P$ and $f_C$ are functions of (chronological) age, period (ie, year of incidence) and cohort (ie, year of birth, approximated by year of diagnosis minus age at diagnosis), respectively. These functions are taken to be natural cubic splines as they offer greater flexibility and allow us to model more realistic trends and projections than models which use the step function since change occurs smoothly rather than in sudden jumps. The logarithmic and 'power-5' (ie, $g(x)=x^{-5}$) link functions were used as both have been found to offer a good fit to the data.[22]

The period effect is a proxy of the effect of screening, which was set to be 0 prior to the introduction of the screening programme (1988) and for women born before 1924, as these birth cohorts would have not benefited from the protective effect of screening when the programme was first launched. The assumption of setting the period to 0 for years prior to 1988 was confirmed by preliminary analyses where we fitted various models to data for years 1971–1987, included. The model indices showed an age cohort model offered a good fit to the data (online supplementary 1). An age cohort model was also shown to provide a good fit for cervical cancer incidence and mortality data pre-1988.[8]

The impact of screening has been found to differ by age group as screening is more effective at older ages[23–26] and wanes after the last screen.[27] Hence, we modelled screening separately for women aged 20–34, 45–49, 50–64, 65–69, 70–74 and 75–84. To capture the effect of screening in different age groups, our modified model (ie, ACP*A) was defined as follows:

$$\lambda(age, period) = g^{-1}[f_A(b\_age) + f_C(cohort) + f_{P,a-b}(period, age)] \quad (2)$$

where $f_{P,a-b}$ is a smooth function of the year of diagnosis, for each age group (a–b), forced to be 0 for cohorts not invited to screening ($f_{P,a-b}[period, age]=0$ if period ≤1988 and cohort <1924), while b_age is the biological age which was defined using the following formula:

$$b\_age = age + max(0, year\ of\ birth - 1955) * x \quad (3)$$

where x represents the number of week(s) per cohort year (ie, weeks/year; ie, age adjustment) to be added to each birth cohort after 1955 and 'age' is the chronological age in years. Therefore, the estimated biological age is a proxy for social changes in sexual behaviour and HPV exposure in successive generations post-1955.

Social changes and the introduction of oral contraceptive in the 1960s led to changes in sexual behaviour (eg, younger age at first intercourse and greater number of sexual partners) and subsequently a rise in sexually

transmitted diseases in the 1970s.[28] We assume that women who were teenagers in the 1970s (and afterwards) have greater and earlier HPV exposure than women from previous cohorts. Thus, in our model, we assumed women born after 1955 have been exposed to HPV for x weeks/year longer than women born before 1955. For example, using a 5 weeks/year adjustment, by 2001 women born in 1975 (chronological age=26) are expected to have been exposed to HPV for approximately two additional years, and therefore have a risk equivalent to women aged 28 born before 1955.

To identify the best model, we compared our modified model (ACP*A) with simpler ones; such as age (A), age+drift (Adrift), age-period (AP), age-cohort (AC) and standard age-period-cohort (APC). We used the pseudo-$R^2$ and Akaike information criterion (AIC) rather than formal significance testing because, with over 100 000 events, even very small deviations from the model may be statistically significant. We conducted a series of validation analyses to identify the age adjustment which best approximated the data. Validation was done in line with previous work exploring cancer incidence in the Nordic countries.[22] The validation used the observed data from 1971 to 2001 to predict incidence rates up to 2013. To identify the best age adjustment, we used a modified Pearson $\chi^2$ statistic (online supplementary 2) to assess how close the projected values were to the observed ones across the estimated years (ie, 2002–2013) using several weeks/year age adjustments (ie, x=0–12) in equation 3.

In all models, we fixed the future period effect to be the same as the one in the last year of observed data (ie, 2013). The cohort effect was allowed to change very slightly for those born after 1981 compared with the 1981 cohort.[29] This was done as we are not expecting past trends to continue into the future and we are unsure how changes in sexual behaviour and HPV vaccination will affect future birth cohorts. Analyses were carried out in Stata v.15.

### Patient and public involvement
No patients or the public were involved in the design or conduct of this study. However, the request to model future cervical cancer rates came from Jo's Cervical Cancer Trust.

### RESULTS
Model fit indices showed that our modified model (ACP*A) offers the best fit to the data using either link function (table 1). Both the log and power link functions offered a good fit to the data; however, the results presented in the rest of the manuscript are based on the model using the log link function as it allows for clear interpretation of the relative risks (RR) associated with the separate effects. The conclusions did not differ depending on the link function used to model the data (online supplementary 3).

**Table 1** Model fit indices for the observed data using the log and power link functions

| | | Log link | | Power link | |
|---|---|---|---|---|---|
| | df | AIC | Pseudo-R²* | AIC | Pseudo-R²* |
| A | 2668 | 99.4 | 0 | 99.1 | 0 |
| Adrift (in cohort) | 2667 | 44.1 | 0.557 | 46.0 | 0.536 |
| AP | 2664 | 37.0 | 0.628 | 39.6 | 0.600 |
| AC | 2661 | 21.0 | 0.789 | 20.5 | 0.793 |
| APC | 2657 | 14.1 | 0.856 | 14.5 | 0.854 |
| ACP*A | 2637 | 10.6 | 0.894 | 10.8 | 0.892 |

*The age-only model is treated as the null model to calculate the pseudo-$R^2$.
AIC, Akaike information criterion.

Moreover, using the best fitting model (ie, ACP*A) the validation analyses showed a 2–7 weeks/year age adjustment offered a good fit to the data with greater adjustments (ie, 5–7 weeks/year), fitting the data best when projections were longer into the future (ie, 6+ years; online supplementary 2). Thus, we used 5 weeks/year for our projections.

Figure 1 shows the observed data (dots) together with projections using chronological age (short-dash dot) compared with biological age with several adjustments using the log function. The greyed area shows the potential range of estimated rates based on different biological age adjustments (ie, 2–7 weeks/year), with higher rates associated with a 2 weeks/year adjustment and lower rates obtained using a 7 weeks/year adjustment. The solid line is for the selected 5 weeks/year adjustment. The model provided a good fit to the observed data using either biological or chronological age; however, future projections differ depending on the age adjustment used. The model, which failed to account for earlier exposure to HPV (short-dash-dot line), estimated future rates (2014–2030) higher than those observed in 1988. This was particularly true for women aged 35–64.

Using a 5 weeks/year adjustment, we calculated the ASR (European standard population) for observed rates in 1972, 1992 and 2012 and estimated rates for 2032 as 3-year averages for all ages combined; for example, rates for 2012 were obtained by averaging rates between 2011 and 2013. We found the observed rates had dropped from 22 to 17 per 100 000 between 1972 and 1992 (−22% percentage change) and had further decreased to 13 per 100 000 in 2012 (percentage change −27%); however, our projections showed rates are expected to increase to 17% from 2012 to 2032 (+39% percentage change) (ASRs across all ages are presented in online supplementary 4).

The figure also depicts rates estimated under a 'no-screening' scenario (ie, counterfactual baseline). These estimates were obtained by only combining the age and cohort effects ($f_A$ and $f_C$), previously estimated from the full model, using a 5 weeks/year age adjustment

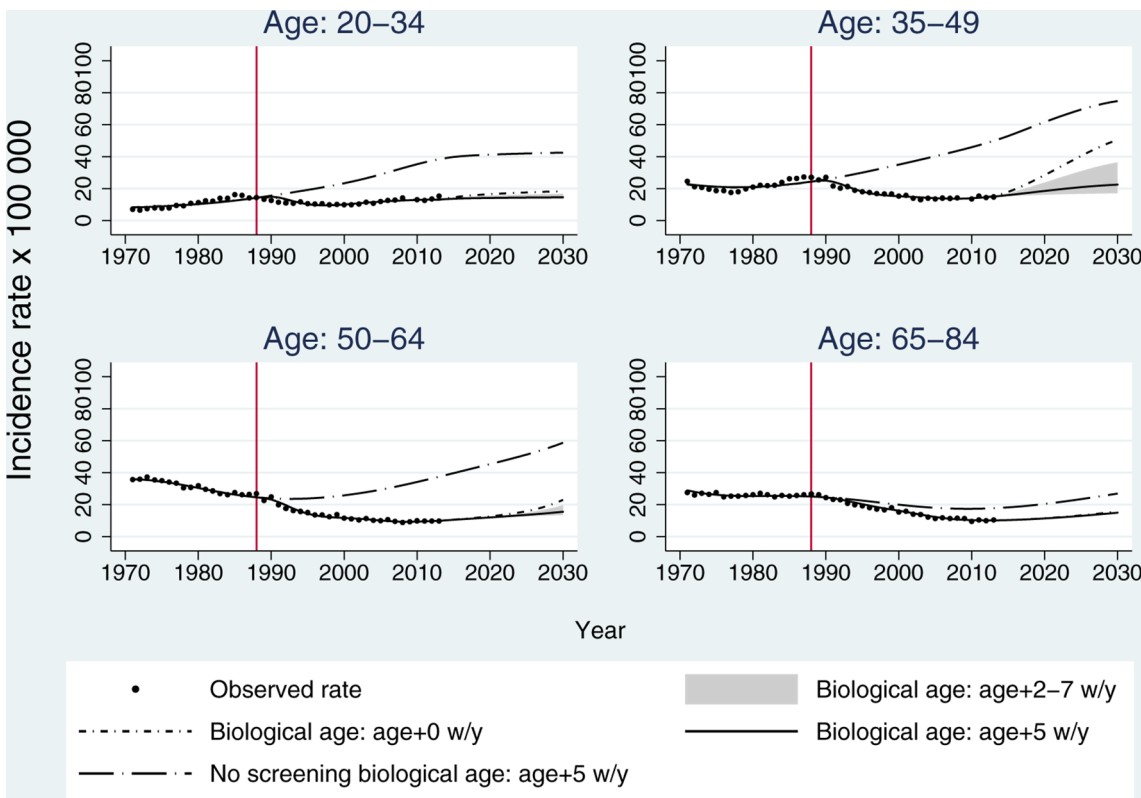

**Figure 1** Trends and projections based on our modified APC model (using the log link function) across different age groups. The solid circles represent the observed data. The estimated rates for the observed period are for chronological age (short-dash dot) and biological age using a 5 weeks/year adjustment (solid line). The greyed area shows the potential range of estimated future rates based on the chronological age using 2 weeks/year and 7 weeks/year age adjustments (biological age). The rates estimated under the no-screening scenario are represented by the long-dash dot line. APC, age period cohort.

(long-dash dot line) for biological age. We estimated that by 2013, approximately 65 000 cases (48%) of the expected cancers had been prevented since the introduction of screening in 1988. Similarly, using the power link function, the estimated number of prevented cancers was over 55 000 (44%). The results showed that, under the no-screening scenario, ASR would have been three-fold higher in 2013 using the log link: observed ASR 12.8 (95% CI 12.3 to 13.3) versus no-screening scenario 37.9 (95% CI 36.4 to 39.3). We also found this will still be the case in 2030 assuming the current scenario continued. In 2013, incidence rates in the no-screening scenario were estimated to be higher across all age groups compared with the observed rates (table 2). As to be expected, the estimated rates under the no-screening scenario were

lower when using the power link compared with the log link function; however, both models showed the protective effect of screening on incidence rates.

### Screening effect

We explored the effect of 'screening' (ie, period effects) across different age groups. Figure 2 shows the RRs associated with the period effect for each age group plotted over the year of diagnosis. The figure shows RRs obtained using chronological age compared with biological age with a 5 weeks/year adjustment. The biological age adjustment does not affect the results for women aged 50+ based on the model assumptions. Overall, screening reduces the RR associated with cancer incidence. This protective effect is more pronounced in women aged

| Table 2 | Incidence rates in 2013 by age group | | |
|---|---|---|---|
| | **Observed data** | **No screening—log link** | **No screening—power link** |
| 20–34 | 14.8 (13.8–15.9) | 34.7 (32.3–37.1) | 29.1 (27.1–31.1) |
| 35–49 | 14.6 (13.6–15.6) | 49.6 (46.2–53.0) | 41.0 (38.2–43.9) |
| 50–64 | 9.8 (8.9–10.7) | 37.9 (34.5–41.3) | 34.5 (31.5–37.6) |
| 65+ | 10.2 (9.2–11.1) | 18.9 (17.1–20.7) | 18.4 (16.7–20.2) |

Observed rates vs estimated rates under the no-screening scenario using a 5-year age adjustment and the log and power link functions.

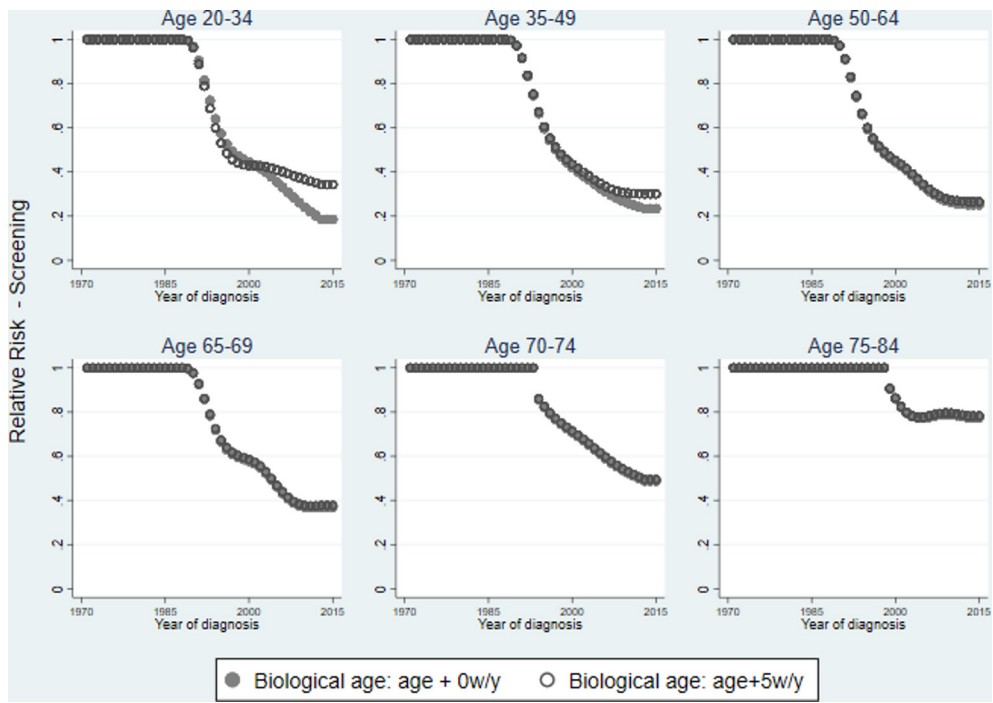

**Figure 2** Relative risk of the period effect by age group for chronological age (solid circle) and biological age using a 5 weeks/year age adjustment (hollow circle).

35–64 (vs <35), but it can still be observed among women aged 65–74 (and even 75–84), although steadily waning, as shown in figure 2.

## DISCUSSION

Our results showed an age-cohort (AC) model could fit incidence rates well for years pre-1988, which indicates the period effect is a proxy for the introduction of screening. Our modified APC model successfully captured past trends and efficiently estimated future incidence rates. Thus, our study corroborates the effectiveness of screening at reducing cervical cancer incidence across all ages.[7–10] Franceschi and Vaccarella estimated screening had prevented more than 30 000 cancers between 1983 and 2007, and we estimated approximately 65 000 were prevented between 1988 and 2013. This increase in the number of prevented cancer is driven by the fact we have six additional years of data. The model showed the period effect, capturing screening, was greater among women aged 35–64 (vs <35), potentially as they have attended several screening rounds. Moreover, it captured the protective effect of screening in older women (65+). Indeed, we observed a waning 'protective' effect which is still considerable at age 74. This is in line with previous results by Castañon and colleagues[27] based on individual exposure to screening (rather than trends). The data suggest there is an effect of screening among women aged 65–69 in the first years after the introduction of the screening programme. This observation is supported by data, which indicate women in this age group were undergoing (opportunistic) screening.[30]

We quantified the effectiveness of the screening programme by comparing observed rates with rates estimated under a no-screening scenario. The results suggest that, had screening not been introduced, overall incidence rates would be threefold higher, which is in line with previous results.[31] Overall, our results confirm findings from previous cohort and case–control studies while using population data, which are readily available.[24] In spite of the protective effect of screening, rates of cervical cancer are projected to increase over time, especially among women aged 35–64, potentially due to increased incidence of HPV infection and falling screening attendance.[32] These results should, however, be interpreted with caution for two reasons. First, they rely on the assumption that the period effects are due to screening, and second we assume that the current scenario will continue. In reality, we are expecting HPV vaccination and the imminent introduction of HPV screening to have a further protective effect on women.[15 32] The counterfactual baseline obtained under the no-screening scenario can be used to quantify the effect of vaccination and HPV screening in the future.

Our results indicate the importance of capturing earlier and prolonged HPV exposure, together with changes in exposure intensity, when estimating future cervical cancer incidence. The assumption that women are exposed to HPV at an earlier age in more recent cohorts is supported by changes in sexual behaviour (eg, earlier age at first intercourse) captured by data from National Surveys of Sexual Attitudes and Lifestyles 3.[14] Failing to account for changes in age of first exposure to HPV leads to unrealistic future rates, akin to those observed in countries

where screening is not systematically offered.[10] This is particularly noticeable among women aged 35–64 rather than younger women (<35). The impact of modelling earlier exposure is minimal among women under the age of 35 since these observations dictate the model fit.

In a recent study, predicted cervical cancer rates for 2035 were higher than prior to 1988, especially among women aged 35–64.[11] These results may be due to the use of a standard APC model, which only captures increased intensity of exposure that is expected to last throughout a woman's lifetime. Moreover, we noticed the standard APC model failed to capture the differential effect of screening by age group in our data . Indeed, using chronological age, screening appeared to have a greater protective effect among younger women (<35) than those aged 35–64. This is counter to previous evidence which found screening to be more effective in older women (40+ vs <40).[23–26] These observations further support the need to consider cohort effects that account for changes in age at first exposure in addition to changes in the proportion of the population exposed, already captured by conventional APC models.

The main strength of our analysis is the use of data broken down by single year of age and year of diagnosis over a 40-year period, which permits predictions using modelling; however, there are some limitations. A typical limitation of APC analysis is that predicted rates are dependent on the model's assumptions, but our validation analyses suggest our assumptions are valid under the current scenario. Additionally, our study does not model the effect of HPV vaccination on future incidence rates in younger cohorts. In the UK, HPV vaccination of girls aged 12–13 years started in 2008 and a catch-up programme ran for women up to the age of 18 for the following 2–3 years. Vaccination uptake has been estimated to be around 80% among eligible girls.[33] However, the counterfactual no-screening scenario baseline from our model was used in a microsimulation study to evaluate the appropriate screening intensity in HPV vaccinated and unvaccinated women.[16] This counterfactual baseline can also be used as a reference to quantify the effect of HPV vaccination and screening in the future. We interpret the reduction in incidence rates being driven by the introduction of the screening programme as indicated by decreasing the RR of the period effect. It is, nonetheless, possible that the period effect is capturing other factors; however, such period reductions are not observed in Eastern European countries, where screening is not adequately implemented, and therefore we believe the reduction is driven by screening in line with previous work.[7 34] Our incidence rates are calculated for the overall population and we did not exclude women who have undergone hysterectomy from the denominator. However, in England and Wales hysterectomy prevalence is low, with a rate of 10.3% among women aged 25–64 (1991–995), and the extrapolated hysterectomy prevalence appears stable up to 2030.[35] Therefore, we are confident our estimated trends closely mirror the trends in cancer incidence in women with a cervix, even though we acknowledge the rates in older women with a cervix will be somewhat higher than the rates in all older women.

## CONCLUSIONS

Using a modified APC model on cancer registry data, we confirmed the protective effect of screening with this effect lasting for over 10 years after the last screening invitation. Our results support the epidemiology of cervical cancer in younger cohorts being affected by earlier onset of sexual behaviour, as reported in previous studies,[14 28] and not just the greater number of sexual partners. We found that failing to account for changes in age of first HPV exposure leads to unrealistic future estimates, and therefore our modified approach should be used when modelling incidence of cervical cancer. Moreover, the use of our model is not restricted to cervical cancer but could be implemented by researchers studying other diseases (eg, lung cancer) where age at exposure, as well as intensity of exposure, may determine future age-specific incidence.

**Contributors** FP was responsible for study design and data analysis and drafted the paper. PS was responsible for study concept and design and contributed to editing of the paper.

**Funding** The research was supported by a grant from Cancer Research UK (C8162/A16892).

**Competing interests** None declared.

**Patient consent for publication** Not required.

**Provenance and peer review** Not commissioned; externally peer reviewed.

**Data sharing statement** The data are available and can be obtained from Public Health England through the Office for Data Release (ODR). The authors can provide data specification for the data request.

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
