## [Reviewer comments · BMJ Open]

This paper was submitted to a another journal from BMJ but declined for publication following peer review. The authors addressed the reviewers' comments and submitted the revised paper to BMJ Open. The paper was subsequently accepted for publication at BMJ Open.

(This paper received three reviews from its previous journal but only two reviewers agreed to published their review.)

ARTICLE DETAILS

TITLE (PROVISIONAL)	The impact of screening on cervical cancer incidence in England: a time trend analysis
AUTHORS	Pesola, Francesca; Sasieni, Peter

VERSION 1 – REVIEW

REVIEWER	Hutch Sripilung Prince of Songkla University Thailand
REVIEW RETURNED	24-Sep-2018

GENERAL COMMENTS	Title: The impact of screening on cervical cancer incidence in England: a time trend analysis Strengths and limitations of this study • Individual level data for cervical cancer incidence over a 40-year window.• First study to use a modified age period cohort (APC) model able to capture generational changes in the prevalence of human papillomavirus (HPV) as well as changes in age at first exposure.• The model allows researchers and policy makers to estimate the effectiveness of screening as well as the effect of HPV vaccination and screening in the future by obtaining a counterfactual baseline.• Our revised model could be used to study other diseases where age at exposure and intensity of exposure affect future age-specific incidence (e.g., lung cancer).• One limitation, typical of APC-based analyses, is that estimates are dependent on the assumptions of the model. Objectives: To explore past trends and estimate future incidence rates of cervical cancer, we developed a modified APC model implicitly able to capture birth-cohort changes in HPV exposure both in terms of changes to cumulative exposure (by age 35) and changes to the age at which women are infected. The latter is the innovative feature of our approach. An important output of our model is that it also allows us to estimate rates under a scenario
---

	where no screening is available (i.e. counterfactual baseline). This counterfactual baseline can be used to quantify the protective effect of screening and the effect of HPV vaccination and screening in future cohorts. Methods: Setting: England Cases: cervical cancers in women aged 20-84 diagnosed between 1971 and 2013 in England. The population estimates (women aged 20-84) for 1971-2014 and projections to 2030 were obtained from the Office for National Statistics. A few cases were excluded with some reasons. APC model APC models, assuming a Poisson distribution used. The statistical method used was described. The authors compared our modified model (ACP*A) with simpler ones (vs. A, Adrift, AP, AC and standard APC) to identify the best model. (At the end of the methods and appendix 1) In all models, we fixed the future period effect to be the same as the one in the last year of observed data (i.e. 2013) and geometric damping was applied to birth cohorts after 1981. This was done as we are not expecting past trends to continue into the future and we are unsure how changes in sexual behaviour and HPV vaccination will affect future birth cohorts Results: Model fit indices showed that our modified model (model VI) offered the best fit to the data. Overall, screening reduces the relative risk associated with cancer incidence. This protective effect is more pronounced in women aged 35-64 (vs. <35) but it can still be observed amongst women aged 65-74 (and even 75-84). We found that failing to account for changes in age of first HPV exposure leads to unrealistic future estimates and, therefore, our modified approach should be used when modelling incidence of cervical cancer. Questions: 1. Line 40: incidence —> incidence The following are my curiosity. If you mention these aspects in the manuscript, international readers could appreciate these information so that we can compare in the future the change in incidence of cervical cancer and HPV infection across countries. 2. Do you have any idea on type of HPV vaccination in England? Who were the target of HPV vaccine: girls or both boys and girls, aged? 3. HPV vaccination is voluntary or compulsory? What is the acceptance among young people and their parents?
--	---

REVIEWER	salvatore vaccarella international agency for research on cancer
REVIEW RETURNED	01-Oct-2018

GENERAL COMMENTS	The article from Pesola and Sasieni aims to assess the impact of screening on cervical cancer incidence in England and to project
---

trends of cervical cancer up to year 2030. The article is well written, clear and introduces into the APC model an interesting feature, ie the possibility to account for the fact that cancer may occur earlier in younger generations because of changes in sexual practices and earlier exposure to HPV. The study confirms the beneficial impact of organized cervical cancer screening programmes.

Comments:

The authors say that “rates of cervical cancer have been decreasing since the introduction of the screening programme in 1988” but decreases may have started earlier due to other reasons, including opportunistic screening, lower fertility rates, etc. For instance, mortality progressively declined much earlier than in 1988 in England. In general, caution should be used to attribute the entire decline to organized screening programmes.

Similarly, period effects may capture all changes that occurred across all ages, not only those relative to organized screening and the assumption in the model of 0 period effect prior to 1988 could be better justified.

Their model estimated that between 1988 and 2013 about 65000 cervical cancer cases have been prevented by screening in England. This estimates is somehow much bigger in absolute terms from that previously estimated by Franceschi and Vaccarella, Cancer Epidemiology 2015 that in their analysis suggested that >30,000 cases (approximately 35% of expected CC cases) may have been prevented by screening programmes between 1983 and 2007 for the whole UK.

It is not completely clear whether the no-screening scenario was estimated, by fitting a different age-cohort model (but in this case, period effects may still be captured by the age and cohort parameters) or by using the age and cohort parameters previously estimated from the full model?

It is always tricky to use interaction terms with APC as the 3 main parameters are interrelated (which may lead to non-identifiability issues and when using step-functions to a saturated model), so to some extent a period-age effect interaction could capture, and be considered a, cohort effects. Could that be clarified?

Also, could you please explain how you have made forecasts for future periods and, in particular, how did you accounted for the fact that by nature a log-linear model will produce exponentially increasing rates, even for instance in the case when the projected (time) parameters would be flat in the log-scale. The use of the standard log link function implies an exponential growth that might lead to a substantial overestimation of the forecasts. To overcome this problem, some authors proposed to change the link function, this is also implemented and commonly used in the Nordpred R-package.

In general, it would important to remember that the effects of the model are ‘interpreted’, for instance, as the screening impact for period effect, as the HPV prevalence in different generation of women for cohort effects even though these variables are not

	directly modeled. This should also be true when interpreting the 'age at which women are infected', a variable not directly modeled.
--	--

VERSION 1 – AUTHOR RESPONSE

Reviewer: 1

1. *Line 40: incidence —> incidence*

We thank the reviewer for spotting this typo. This has now been amended (page 11).

The following are my curiosity. If you mention these aspects in the manuscript, international readers could appreciate these information so that we can compare in the future the change in incidence of cervical cancer and HPV infection across countries.

2. *Do you have any idea on type of HPV vaccination in England? Who were the target of HPV vaccine: girls or both boys and girls, aged?*
3. *HPV vaccination is voluntary or compulsory? What is the acceptance among young people and their parents?*

We now include additional information re. HPV vaccination in the UK in the introduction (page 11)

Reviewer: 2

1. *The authors say that “rates of cervical cancer have been decreasing since the introduction of the screening programme in 1988” but decreases may have started earlier due to other reasons, including opportunistic screening, lower fertility rates, etc. For instance, mortality progressively declined much earlier than in 1988 in England. In general, caution should be used to attribute the entire decline to organized screening programmes.*

We now modified the text in the abstract to read “Rates of cervical cancer incidence have been decreasing since the introduction of screening but are projected to increase in the future under the current scenario”. This is to indicate that it not entirely due to the introduction of the national screening programme. We also added this consideration in the discussion (page 11)

2. *Similarly, period effects may capture all changes that occurred across all ages, not only those relative to organized screening and the assumption in the model of 0 period effect prior to 1988 could be better justified.*

We now report model fit indices for the various models applied to data pre-1988 (1971 to 1987, included). These results indicate that an age-cohort model offers a good fit to data pre-1988, which supports the assumption of setting the period to be equal to 0 prior to 1988. This is also discussed in the paper to support our assumption (page 6). This is in line with some previous work where historic incidence and mortality data were modelled using APC models and an AC model was found to fit the data well pre-1988 (Sasieni & Adams, 2000). Moreover, the figure in Supplement 4 indicates that cervical cancer incidence rates were fairly stable between 1972 and 1988 whereas they started to decrease from 1988, which coincides with the introduction of the screening programme.

3. *Their model estimated that between 1988 and 2013 about 65000 cervical cancer cases have been prevented by screening in England. This estimates is somehow much bigger in absolute terms from that previously estimated by Franceschi and Vaccarella, Cancer Epidemiology 2015 that in their analysis suggested that >30,000 cases (approximately 35% of expected CC cases) may have been prevented by screening programmes between 1983 and 2007 for the whole UK.*

When we restrict our analysis to the years between 1983 and 2007, we find that, using the log link, just >37000 cancers (~35%; vs. power link: approximately 33000, ~31%) may have been prevented by the screening programme in line with the paper by Franceschi and Vaccarella. Thus, the absolute value reported in our paper is greater than the one in Franceschi and Vaccarella’s paper due to additional 6 years of data. We now mention this in the discussion (page 9). The difference in the

results may be due to the fact we modelled the impact of screening in different age groups as there are different trends in younger women.

4. It is not completely clear whether the no-screening scenario was estimated, by fitting a different age-cohort model (but in this case, period effects may still be captured by the age and cohort parameters) or by using the age and cohort parameters previously estimated from the full model?

The rates under the no-screening scenario were obtained by combining the age and cohort parameters previously estimated from the full model. This is now clarified on page 8 as: “These estimates were obtained by only combining the age and cohort effect (fA and fC), previously estimated from the full model, with a 5-weeks/year age adjustment (long-dash dot line).”

5. It is always tricky to use interaction terms with APC as the 3 main parameters are interrelated (which may lead to non-identifiability issues and when using step-functions to a saturated model), so to some extent a period-age effect interaction could capture, and be considered a, cohort effects. Could that be clarified?

The reviewer is correct in the fact that ‘cohort = period – age’ means that an age*period interaction could also be interpreted as an age*cohort interaction; however, we find the latter may be difficult to interpret. The proposed A*P term is to allow different screening effect in the various age groups.

6. Also, could you please explain how you have made forecasts for future periods and, in particular, how did you account for the fact that by nature a log-linear model will produce exponentially increasing rates, even for instance in the case when the projected (time) parameters would be flat in the log-scale. The use of the standard log link function implies an exponential growth that might lead to a substantial overestimation of the forecasts. To overcome this problem, some authors proposed to change the link function, this is also implemented and commonly used in the Nordpred R-package.

We opted for the log link function as it allows for a clear interpretation of the relative risks associated with the separate effects. Following the reviewer’s comment, we compare the results from the models using the log and ‘power of 5’ link functions. We find that both the log and power link functions offered a good fit to the observed data and produce similar results for the projected rates up to 2030 (figure below). The main difference is observed in the estimated rates under the no-screening scenario, especially amongst women aged 35-49. Following these considerations, we are confident that presenting the results based on the log link function is sufficient as it allows for a clear interpretation. We now include figure 1 below in the supplementary materials.

Figure 1. Observed and estimated rates obtained using the log and power link functions and a 5-week/year age adjustment for biological age

7. In general, it would important to remember that the effects of the model are 'interpreted', for instance, as the screening impact for period effect, as the HPV prevalence in different generation of women for cohort effects even though these variables are not directly modeled. This should also be true when interpreting the 'age at which women are infected', a variable not directly modeled.

In the paper, we now stress that the period effect and cohort effects are proxies for screening and changes in sexual behaviour, respectively (e.g. pages 4 and 5). We now also make it clear that the "estimated biological age is a proxy for social changes in sexual behaviour and HPV exposure in successive generations post 1955" (page 6). We also stress the fact that it is not possible to prove causality from changes in trends but that the simplest interpretation of what we observe is that the period effects capture screening effects.

We hope that the changes we have made to the manuscript meet with your approval and that the revised manuscript is now suitable for publication.

VERSION 2 – REVIEW

REVIEWER	Salvatore Vaccarella International Agency for Research on Cancer
REVIEW RETURNED	16-Nov-2018
GENERAL COMMENTS	nice paper, ready to be published